# Exploring the relationship between trunk flexibility and arterial stiffness measured by pulse wave velocity: A systematic review and meta-analysis

**Iván Cavero-Redondo**[1,2], **Helder Fonseca**[3,4], **Iris Otero-Luis**[1]*, **Lucimere Bohn**[3,4,5], **Carla Geovanna Lever-Megina**[1], **Nerea Moreno-Herraiz**[1], **Alicia Saz-Lara**[1]

**1** CarVasCare Research Group, Facultad de Enfermería de Cuenca, Universidad de Castilla-La Mancha, Cuenca, Spain, **2** Facultad de Ciencias de la Salud, Universidad Autonoma de Chile, Talca, Chile, **3** Research Center in Physical Activity, Health and Leisure (CIAFEL), Faculty of Sports, University of Porto, Porto, Portugal, **4** Laboratory for Integrative and Translational Research in Population Health (ITR), Porto, Portugal, **5** Faculty of Psychology, Education and Sport, Lusófona University of Porto, Porto, Portugal

* iris.otero@uclm.es

**Data Availability Statement:** All data are in the manuscript and/or Supporting information files.

## Abstract

### Background

As individuals age, the risk of cardiovascular disease (CVD) increases, largely due to progressive stiffening of the arteries. This relationship underscores the critical need to monitor arterial stiffness as a predictor of CVD outcomes. While aerobic exercise has demonstrated benefits for vascular health, the influence of flexibility, particularly trunk flexibility, on arterial stiffness remains underexplored. Thus, this study aimed to analyse the overall relationship between trunk flexibility and arterial stiffness across different age groups (young, middle-aged, and older adults) and according to sex.

### Methods

A systematic review and meta-analysis were conducted following the MOOSE and JBI Manual for Evidence Synthesis on systematic reviews of etiology and risk guidelines. Searches in PubMed, Scopus, and Web of Science identified studies investigating the association between trunk flexibility and arterial stiffness measured by pulse wave velocity. Data extraction, quality assessment, and statistical analyses were performed following predefined criteria.

### Results

Five studies involving 2797 participants were included. Poor trunk flexibility was associated with increased arterial stiffness compared to high flexibility (pooled standardized mean difference = -0.27, 95% CI: -0.39, -0.14), with substantial heterogeneity observed. Subgroup analyses by sex and age revealed significant associations predominantly in men and older individuals. Sensitivity analyses confirmed the robustness of the findings, and meta-

**Funding:** The author(s) received no specific funding for this work.

**Competing interests:** The authors have declared that no competing interests exist.

regression models showed no significant differences according to age, BMI, or blood pressure. No evidence of publication bias was found.

## Conclusion

Poor trunk flexibility is linked to elevated arterial stiffness across diverse demographic groups, highlighting its potential as a surrogate marker for cardiovascular health. Physiological mechanisms involving connective tissue integrity and neural regulation may underpin this relationship. Understanding the role of flexibility in arterial health could inform targeted interventions to mitigate age-related increases in arterial stiffness and reduce cardiovascular disease risk. However, further research is needed to validate these findings and explore potential sex-specific differences.

## Introduction

Cardiovascular disease (CVD) is the primary cause of mortality worldwide, and age serves as a pivotal determinant of cardiovascular health [1]. Approximately 80% of deaths among older adults are attributed to CVD [2], with age-related increases in blood pressure and central arterial stiffness significantly contributing to CVD risk [3]. Arterial stiffness, indicative of diminished arterial compliance, is a key marker of vascular health and a predictor of arterial hypertension and negative cardiovascular events [4]. While aging is often associated with immutable arterial stiffening, lifestyle behaviors, including physical inactivity and lack of exercise, may influence this trajectory [5].

Habitual physical activity, particularly aerobic exercise, has been consistently linked to reduced blood pressure and arterial stiffness, thereby mitigating CVD risk [6]. Nevertheless, despite the strong evidence supporting the benefits of high aerobic capacity on vascular health [7], other components of physical fitness also seem to significantly influence arterial stiffness [8–10]. In particular, the impact of flexibility, specifically trunk flexibility, on arterial stiffness is a topic of ongoing investigation [11]. Cross-sectional studies have identified associations between poor trunk flexibility and elevated arterial stiffness, independent of other fitness components, suggesting potential physiological mechanisms underlying this relationship [12–15]. Understanding how flexibility impacts arterial stiffness could offer novel insights into preventive strategies and interventions for reducing CVD burden.

The sit-and-reach test is a simple and widely used method to assess trunk flexibility, particularly the flexibility of the lower back and hamstring muscles. It is commonly used in both clinical and fitness settings to evaluate an individual's range of motion and is considered a general indicator of overall flexibility [16]. Furthermore, pulse wave velocity (PWV) is a non-invasive measure of arterial stiffness, which is a key indicator of vascular health [17]. Specifically, brachial-to-ankle PWV (baPWV) measures the stiffness of peripheral arteries, while carotid-to-femoral PWV (cfPWV) focuses on central arterial stiffness, particularly the aorta [18]. These measurements are clinically relevant as higher PWV values are associated with increased cardiovascular risk and are used to predict adverse cardiovascular events [19–21].

This systematic review and meta-analysis aimed to investigate the physiological association between trunk flexibility and arterial stiffness, as assessed by PWV. By synthesizing data from cross-sectional studies, we aimed to analyse the overall relationship between trunk flexibility and arterial stiffness across different age groups (young, middle-aged, and older adults) and

according to sex. This comprehensive analysis sought to elucidate the role of trunk flexibility in arterial health, providing potentially valuable insights for the development of targeted interventions to mitigate age-related increases in arterial stiffness and reduce the risk of cardiovascular disease among older adults.

## Methods

In conducting this systematic review, we adhered to the guidelines set forth by the MOOSE statement to ensure comprehensive and accurate data synthesis [22] (S1 Checklist). Additionally, this systematic review was conducted following the guidelines outlined in the JBI Manual for Evidence Synthesis on systematic reviews of etiology and risk [23]. This study was registered *a priori* in PROSPERO (registration number: CRD42024529894).

### Search strategy

A systematic literature search was independently conducted by two reviewers (AS-L and IC-R) through three databases—PubMed, Scopus, and Web of Science—from their inception to March 30, 2024. To perform the search, the following free terms combined with Boolean operators (AND, OR) were used following the population, exposure, comparison, outcome, study design (PECOS) strategy [24]: "trunk flexibility", "flexibility", "arterial stiffness", "aortic stiffening", "arterial stiffening", "pulse wave velocity", "PWV", "brachial to ankle pulse wave velocity", "baPWV", "carotid to femoral pulse wave velocity", and "cfPWV". No time restrictions or other search filters were used. S1 Table in S1 File shows the search strategy for the databases used. Furthermore, previous systematic reviews or meta-analyses were searched, and the references of the included articles were analysed. A final search was performed just before the final analysis to include the most recently published studies.

### Selection criteria

Studies on the association between poor trunk flexibility and high trunk flexibility with arterial stiffness were included in the systematic review and meta-analysis. The inclusion criteria were as follows: (1) population: healthy or apparently healthy subjects of both sexes without comorbidities (older than 18 years); (2) exposure: poor trunk flexibility, as measured by the sit-and-reach test; (3) comparison: high trunk flexibility, as measured by the sit-and-reach test; (4) outcome: arterial stiffness measured by baPWV or cfPWV. PWV is a widely recognised measure of arterial stiffness, with higher values indicating stiffer arteries and increased cardiovascular risk [25]. In the case of cfPWV, a value above 1000 cm/s is generally considered a threshold of significant arterial stiffness, which is associated with an increased risk of cardiovascular events such as myocardial infarction and stroke [26]. In contrast, for baPWV, a threshold of approximately 1400 cm/s is generally considered indicative of significant arterial stiffness [27]; and (5) study design: cross-sectional or baseline data from longitudinal studies. The reason to include cross-sectional studies and baseline data from longitudinal studies was based on the objective of assessing the relationship between trunk flexibility and arterial stiffness at a specific point in time in various populations. Cross-sectional studies are ideally suited to capture this relationship at a single point in time, providing a snapshot of how these variables correlate in different demographic groups [28]. The inclusion of baseline data from longitudinal studies allows the integration of results from studies that, while designed to track changes over time, also provide valuable baseline data comparable to cross-sectional studies. This approach allows for a more comprehensive analysis of existing evidence while maintaining consistency in the temporal assessment of the relationship between trunk flexibility and arterial stiffness. We excluded (1)

review articles, editorials, or case reports; (2) articles that were not written in English or Spanish; and (3) interventional studies.

## Data extraction and quality assessment

Study selection, data extraction, and quality assessment were independently performed by two researchers (AS-L and IC-R), excluding those studies that did not meet the eligibility criteria. Disagreements were resolved by consensus or with the intervention of a third researcher.

The main characteristics of the included studies are summarized in Table 1, which includes information on (1) reference: first author and year of publication, (2) country in which the study data were collected, and (3) population characteristics for poor trunk flexibility and high trunk flexibility: sample size (percentage of women in the total sample size), mean age, mean sit-and-reach values and mean PWV values.

The quality assessment tool for observational cohort and cross-sectional studies from the United States National Institute of Health National Heart, Lung, and Blood Institute [29] was used to assess the risk of bias according to the following domains: quality of the research question, reporting of the population definition, participation rate, recruitment, sample size, appropriateness of statistical analyses, timeframe for associations, exposure levels, ascertainment of the exposure, appropriateness of the outcome measured, outcome blinding of researchers, loss to follow-up, and confounding variables. The overall bias of each study was considered "good" if most criteria were met and there was a low risk of bias, "fair" if some criteria were met and there was a moderate risk of bias, or "poor" if few criteria were met and there was a high risk of bias.

Table 1. Characteristics of the included studies.

| Reference | Country | PWV type | Poor trunk flexibility | | | | High trunk flexibility | | | |
|---|---|---|---|---|---|---|---|---|---|---|
| | | | Sample size (% Women) | Mean age (years) | Sit-and-reach (cm) | PWV (cm/s) | Sample size (% Women) | Mean age (years) | Sit-and-reach (cm) | PWV (cm/s) |
| Gando et al. 2017 [33] | Japan | cfPWV | 99 (76.8) | 49.8±9.4 | 30.2±6.0 | 835.0±164.0 | 206 (72.3) | 49.5±9.6 | 44.9±5.2 | 854.0±140.0 |
| Komatsu et al. 2017 [13] | Japan | cfPWV | Middle-age: 66 (84.8) Older: 35 (74.3) | Middle-age: 58.0±4.0 Older: 69 ±4.0 | Middle-age: 33.0±6.0 Older: 33.0 ±7.0 | Middle-age: 1060.0±183.0 Older: 1198.0 ±220.0 | Middle-age: 64 (84.4) Older: 33(69.7) | Middle-age: 58.0±4.0 Older: 68.0 ±3.0 | Middle-age: 47.0±5.0 Older: 45.0 ±6.0 | Middle-age: 1030.0±158.0 Older: 1092 ±160.0 |
| Nishiwaki et al. 2014 [15] | Japan | baPWV | Young: 206 (49.0) Middle-age: 202 (54.5) Older: 172 (57.0) | Young: 24.5 ±1.0 Middle-age: 49.5±1.0 Older: 69.0 ±1.0 | Young: 34.8 ±0.7 Middle-age: 34.5±0.6 Older: 25.3 ±0.7 | Young: NA Middle-age: NA Older: NA | Young: 204 (48.5) Middle-age: 199 (54.3) Older: 169 (40.2) | Young: 24.0 ±1.0 Middle-age: 59.5±1.0 Older: 68.5 ±1.0 | Young: 48.8 ±0.6 Middle-age: 44.2±0.6 Older: 41.1 ±0.7 | Young: NA Middle-age: NA Older: NA |
| Yamamoto et al. 2009 [14] | Japan | baPWV | Young: 92 (NA) Middle-age: 100 (NA) Older: 61 (NA) | Young: 26.0 ±1.0 Middle-age: 49.0±1.0 Older: 67.0 ±1.0 | Young: 32.0 ±1.0 Middle-age: 31.0±1.0 Older: 26.0 ±1.0 | Young: 1085.0 ±11.0 Middle-age: 1260.0±14.0 Older: 1485.0 ±29.0 | Young: 98 (NA) Middle-age: 104 (NA) Older: 71 (NA) | Young: 26.0 ±1.0 Middle-age: 49.0±1.0 Older: 67.0 ±1.0 | Young: 47.0 ±1.0 Middle-age: 46.0±1.0 Older: 41.0 ±1.0 | Young: 1080.0 ±12.0 Middle-age: 1200.0±12.0 Older: 1384.0 ±24.0 |
| Yoo et al. 2022 [12] | Korea | baPWV | 206 (50.5) | 72.2±4.5 | 2.3±5.8 | 1688.5±338.1 | 410 (49.8) | 70.5±4.0 | 16.9±3.0 | 1611.1±147.4 |

The data are shown as the means ± standard deviations. ba-PWV: brachial-to-ankle pulse wave velocity, cfPWV: carotid-to-femoral pulse wave velocity, NA: not available, PWV: pulse wave velocity.

## Data synthesis and statistical analysis

The DerSimonian and Laird random effects method [30] was used to compute pooled estimates of mean values and their respective 95% confidence intervals (95% CIs) for sit-and-reach and arterial stiffness (baPWV or cfPWV) in poor and high trunk flexibility. In addition, pooled estimates of the standardized mean differences (SMDs) and their respective 95% CIs between poor and high trunk flexibility were calculated for baPWV and cfPWv. Heterogeneity was assessed using the I2 statistic, which ranges from 0 to 100%. According to the I2 values, heterogeneity was considered not important (0 to 30%), moderate (30 to 60%), substantial (60 to 75%), or considerable (75 to 100%) [31]. The corresponding p values were also considered.

Sensitivity analysis (systematic reanalysis removing studies one at a time) was conducted to assess the robustness of the summary estimates. Subgroup analyses were performed for sex and age groups (young, middle-aged and older). Random effects meta-regressions were used to assess whether mean age, body mass index (BMI), systolic blood pressure (SBP), diastolic blood pressure (DBP), baseline sit-and-reach values and baseline PWV values as continuous variables modified the SMD between poor and high trunk flexibility and PWV. Finally, publication bias was evaluated using Egger's regression asymmetry test [32]. A level of <0.10 was used to determine whether publication bias was present.

Statistical analyses were performed using STATA SE software, version 15 (StataCorp, College Station, TX, USA).

## Results

### Baseline characteristics

In total, five studies [12–15, 33] were included in the systematic review and meta-analysis (Fig 1 and S2 File). Four included studies were cross-sectional [12–15], and one was a longitudinal study [33]. Studies were conducted in two countries—four in Japan [13–15, 33] and one in Korea [12]—and were published between 2009 and 2022. A total of 1239 and 1558 subjects were included for poor and high trunk flexibility, respectively (age range 24–72 years). Finally, three studies measured baPWV (12,14,15) and two measured cfPWV [13, 33] for determining arterial stiffness (Table 1). Additionally, S2 Table in S1 File displays information regarding BMI, SBP and DBP in the included studies.

### Quality assessment and potential bias

The overall risk of bias for studies examining the relationship between trunk flexibility and arterial stiffness was good in 60% of studies and fair in 40% of studies (S3 Table in S1 File). Based on the scale assessment items, we were able to identify two main reasons for a fair risk of bias: (i) a sample size justification was not reported, and (ii) whether the researchers were blinded to the exposure status of the participants was not reported. In addition, due to the cross-sectional design of the studies, assessments of exposures over time were not performed.

### Associations between poor and high trunk flexibility and arterial stiffness

The pooled mean sit-and-reach distance for the poor trunk flexibility group was 28.18 cm (95% CI: 25.32, 31.05), and that for the high trunk flexibility group was 42.17 cm (95% CI: 38.84, 45.51) (S1 Fig in S1 File). The pooled mean baPWV for poor trunk flexibility was 1377.87 cm/s (95% CI: 1225.32, 1530.41), and that for high trunk flexibility was 1318.26 cm/s (95% CI: 1197.97, 1438.55) (S2 Fig in S1 File). Finally, the pooled mean cfPWV for poor trunk flexibility was 1029.07 cm/s (95% CI: 823.50, 1234.64), and that for high trunk flexibility was 990.83 cm/s (95% CI: 842.43, 1139.24) (S3 Fig in S1 File).

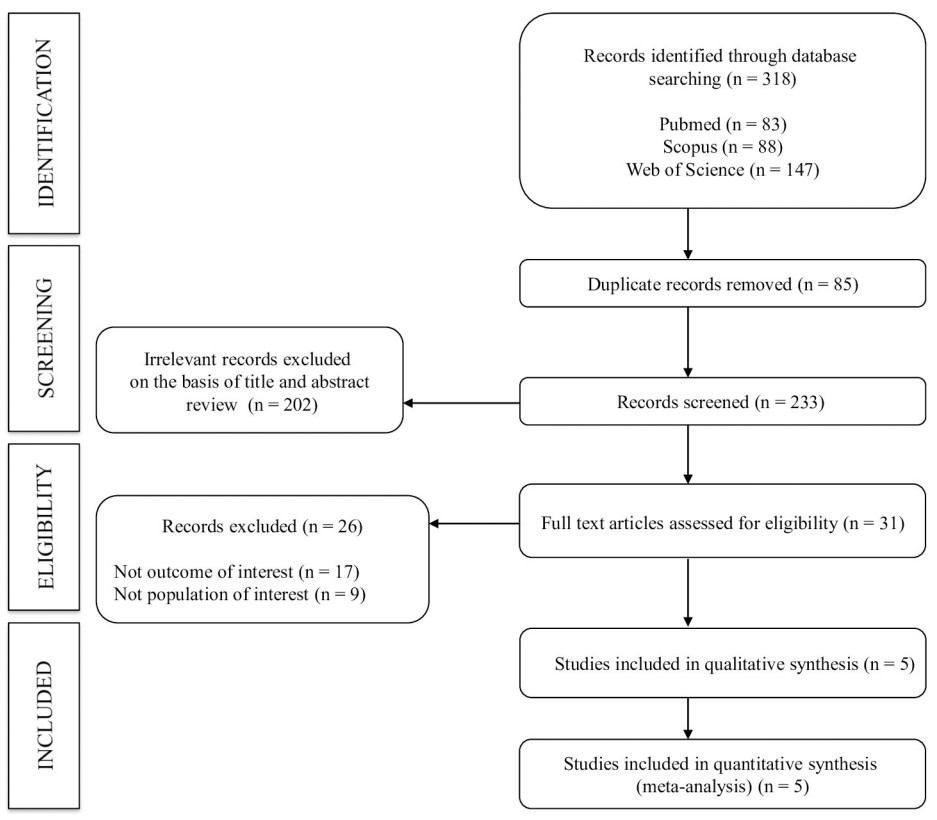

**Fig 1. PRISMA flow chart.**

The pooled SMD of the PWV between patients with poor and high trunk flexibility was −0.27 cm/s (95% CI: −0.39, −0.14), with substantial heterogeneity ($I^2$ = 67.4%); the SMD for the baPWV was −0.30 cm/s (95% CI: −0.42, −0.18), with moderate heterogeneity ($I^2$ = 57.2%); and that for the cfPWV was −0.15 cm/s (95% CI: −0.51, 0.2), with substantial heterogeneity ($I^2$ = 73.2%) (Fig 2).

### Sensitivity analysis, subgroup analyses, meta-regression models and publication bias

The pooled SMD of the PWV for the poor and high trunk flexibility groups was not significantly modified (in magnitude or direction) when data from individual studies were removed from the analysis one at a time.

Subgroup analysis by sex and by age group revealed that only men (−0.40 cm/s, 95% CI: −0.53, −0.28) and older subjects (−0.36 cm/s, 95% CI: −0.50, −0.21) had a statistically significant difference in the SMD of PWV between those with poor and high trunk flexibility (Figs 3 and 4).

Meta-regression models showed that mean age, BMI, SBP and DBP did not modify the pooled SMD of PWV between patients with poor and high trunk flexibility (S4–S7 Figs in S1 File).

Finally, no evidence of publication bias was detected using Egger's test (p = 0.136) (S8 Fig in S1 File).

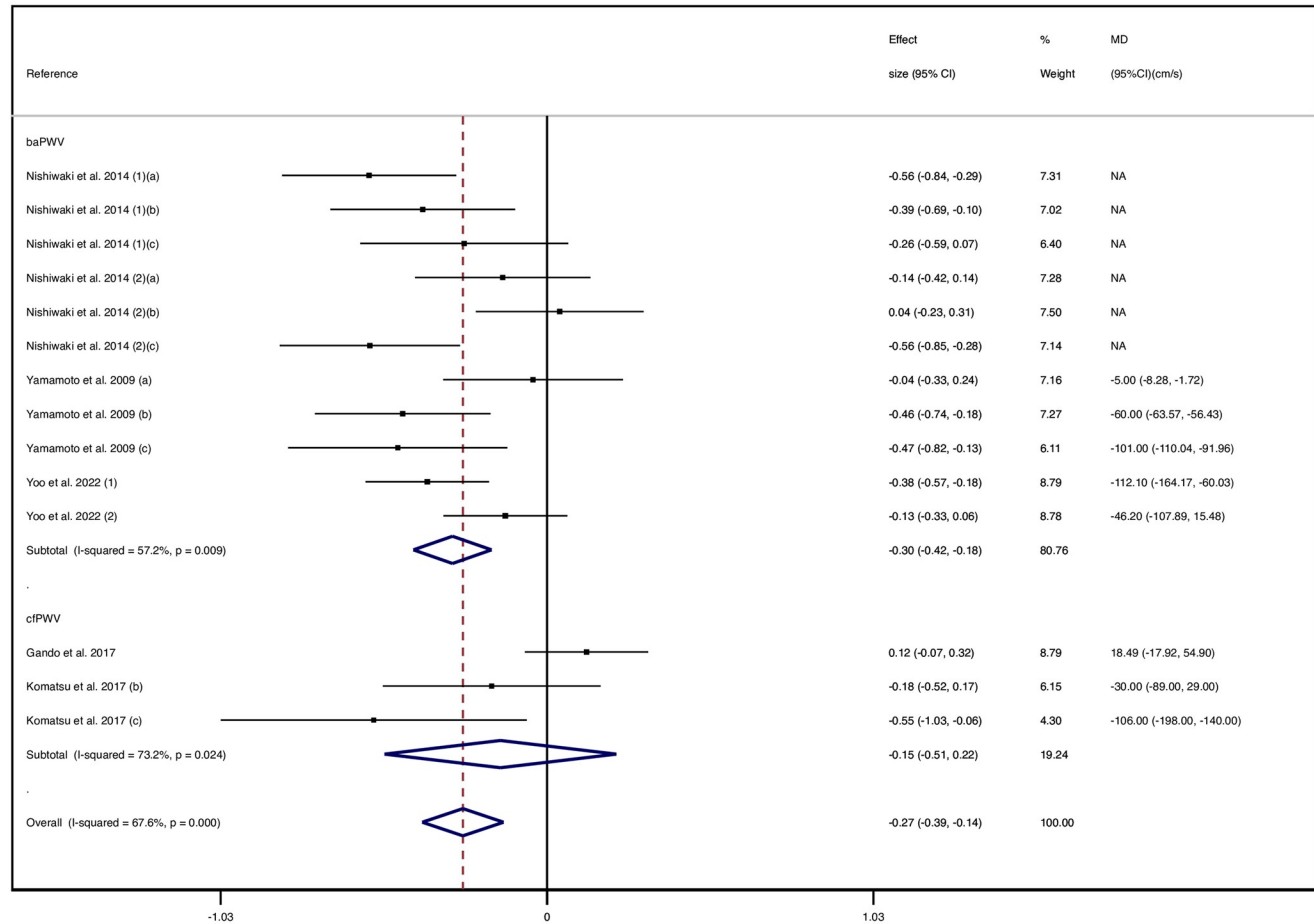

**Fig 2. Forest plot for pooled standardized mean differences in pulse wave velocity between patients with poor and high trunk flexibility.**

## Discussion

Our study synthesized the available data on the relationship between trunk flexibility and arterial stiffness, focusing on different demographic groups. We found that poor trunk flexibility was significantly associated with increased arterial stiffness, as indicated by increased PWV, across various age and sex categories. Notably, this relationship persisted even after adjusting for potential confounding factors such as BMI, SBP and DBP.

Our findings regarding cfPWV align with previous studies demonstrating an inverse correlation between trunk flexibility and arterial stiffness [14, 33]. This association could be explained by the interconnected functions of connective tissue health and vascular elasticity, where both muscle and collagen structures play a crucial role in maintaining arterial compliance, which influence both flexibility and arterial stiffness [34], as well as the effect of similar mechanisms contributing to both connective tissue and artery stiffness, such as inflammation [35] and the formation of advanced glycation products [36].

Additionally, neural regulation of arterial vascular tone may play a role, with habitual stretching exercises potentially reducing arterial stiffness by modulating sympathetic nerve activity [37]. Similarly, our results concerning baPWV corroborate the inverse relationship between trunk flexibility and arterial stiffness [12, 14, 15]. The observed association suggested

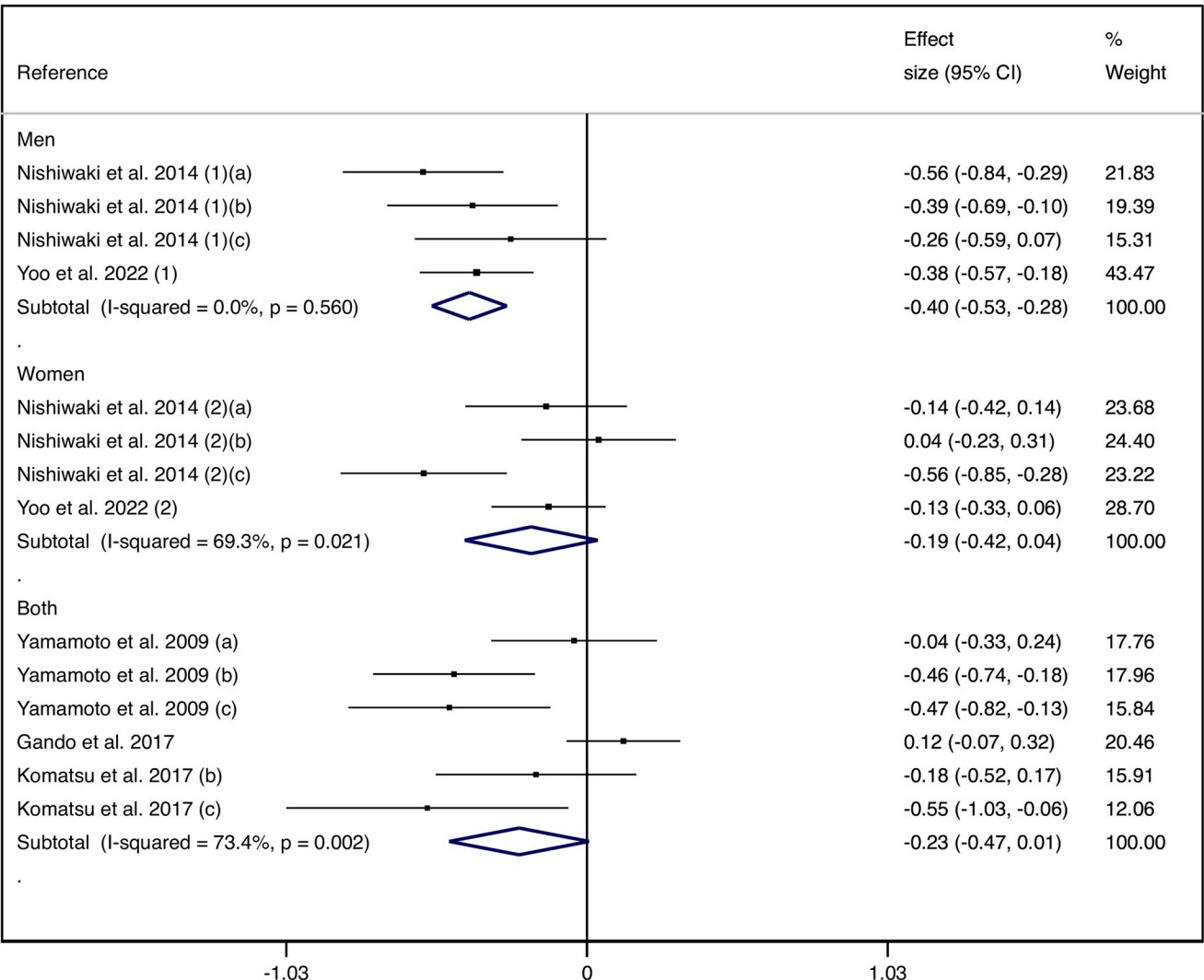

**Fig 3. Forest plot for pooled standardized mean differences in pulse wave velocity between poor and high trunk flexibility groups stratified by sex.**

that flexibility exercises, which promote elasticity in connective tissues, may contribute to maintaining arterial compliance [38]. This finding is consistent with prior research indicating that stretching programs can reduce arterial stiffness, potentially by influencing vascular tone and endothelial function [11]. It is also plausible that the same mechanisms that can contribute to vascular health and endothelial function, thereby reducing arterial stiffness, can also contribute to reduced connective tissue stiffness, such as low systemic inflammation and adequate metabolic and glycemic profiles.

An important limitation to consider is the inherent differences between cfPWV and baPWV measurements. cfPWV is often considered the gold standard for assessing central arterial stiffness, as it directly measures aortic stiffness, which is closely related to cardiovascular events [39]. In contrast, baPWV captures peripheral arterial stiffness, which may have different clinical implications [40]. Previous studies have highlighted that baPWV values are often higher than cfPWV, due to the inclusion of peripheral arteries in the measurement [18]. These differences could influence the interpretation of arterial stiffness and associated cardiovascular

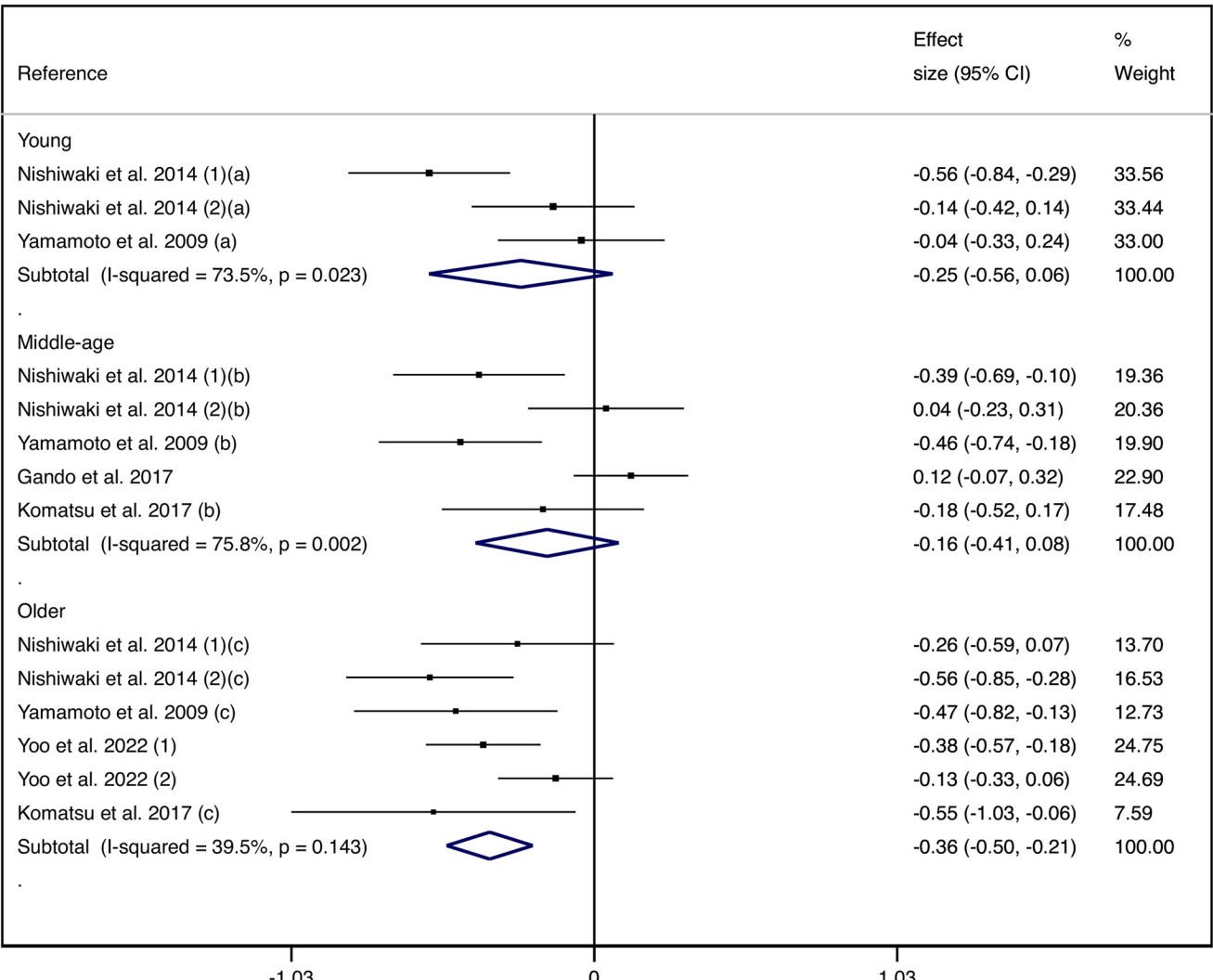

**Fig 4. Forest plot for pooled standardized mean differences in pulse wave velocity between patients with poor and high trunk flexibility by age group.**

risk. Consequently, when comparing or interpreting results, it is crucial to consider these methodological differences to avoid overestimation or misclassification of vascular health status. Future studies should aim to standardise measurements of PWV or adjust for these differences to ensure consistency and accuracy in cardiovascular risk assessment.

Our study revealed significant associations between trunk flexibility and arterial stiffness across different age categories. Specifically, men and older individuals with poor trunk flexibility exhibited stiffer arteries than those with greater flexibility. This finding contrasts with previous findings suggesting no significant relationship between flexibility and arterial stiffness in younger individuals [15]. The age-related increase in arterial stiffness may be attributed to changes in connective tissue composition, neural regulation and metabolic profile, highlighting the importance of considering age-specific factors in cardiovascular health [41].

Considering sex-specific differences in the flexibility–arterial stiffness relationship, men across various age groups showed significant correlations between flexibility and arterial

stiffness; this association was less pronounced in older women. This disparity may be influenced by hormonal factors, with estrogen and testosterone levels potentially modulating arterial compliance [42]. The vasodilatory effects of estrogen and the protective role of testosterone against vascular stiffness may explain these sex-specific differences [43].

Despite the strengths of this systematic review and meta-analysis, including its comprehensive search strategy and adherence to a rigorous methodology, there are some limitations to consider. First, heterogeneity was observed in the pooled SMD, suggesting potential variations across the included studies. This heterogeneity may be attributed to differences in participant characteristics or measurement techniques. Although our subgroup and sensitivity analyses by age, measurement techniques and sex provide some information on sources of heterogeneity, other participant characteristics may also play a role. Variations in physical activity levels, which were not uniformly reported across studies, could influence trunk flexibility and arterial stiffness differently, leading to variability in results. In addition, underlying health conditions, such as hypertension or diabetes, and differences in the specific methods or protocols used to assess trunk flexibility, may further contribute to the observed heterogeneity. These factors underscore the complexity of interpreting associations between trunk flexibility and arterial stiffness, highlighting the need for more standardised assessment protocols and comprehensive participant data in future studies. Second, there were differences in the poor and high trunk flexibility classifications in the included studies. In particular, some studies defined these categories based on population-specific percentiles, while others used absolute cut-off values derived from the distribution of flexibility measures within their samples. These different definitions could introduce variability in the pooled analysis, as what constitutes 'low' or 'high' flexibility could differ between study populations. Furthermore, it is important to note that our meta-analysis was based on the flexibility categories reported by the original studies, and that we did not have direct access to the raw flexibility measures of individual participants. Therefore, caution should be taken when interpreting the results, as variability in definitions could affect the generalisability of our findings. Third, although sensitivity analyses did not alter the overall findings, the possibility of residual confounding cannot be completely ruled out. Fourth, the included studies were carried out mostly in Asian populations, which might reduce the external validity of the findings. Fifth, the lack of detailed information on the overall physical activity levels of participants in the included studies. Although flexibility is a component of fitness, often correlated with other aspects of physical activity, the independence of trunk flexibility from overall activity levels remains unclear [44]. This gap suggests that further research is needed to examine whether trunk flexibility may be an independent predictor of arterial stiffness or whether it is significantly confounded by overall physical activity. Future studies should aim to include comprehensive assessments of physical activity to better understand its role in the observed associations.

For future research, it is essential to control for potential confounding factors that could influence the relationship between trunk flexibility and arterial stiffness. Key confounders to consider include physical activity levels, as regular exercise could independently affect both flexibility [45] and arterial stiffness [7, 37, 46]. In addition, underlying diseases such as hypertension, diabetes and obesity [47, 48], which are known to affect vascular health, should be considered. Lifestyle factors such as diet [49–51], sleep [52, 53] and smoking [54] may also play an important role and should be monitored in future studies to isolate the specific contribution of trunk flexibility to arterial stiffness. Controlling for these variables will help to clarify the independent effects of flexibility and improve the accuracy of predictive models of cardiovascular risk.

## Conclusions

In conclusion, our study underscores the importance of trunk flexibility as a predictor or surrogate of arterial stiffness across different demographic groups. Understanding the physiological and pathological mechanisms underlying this relationship can inform targeted interventions aimed at improving cardiovascular health. Further research is warranted to validate the clinical utility of flexibility assessment in predicting arterial health and to explore potential sex-specific differences in the relationship between flexibility and arterial stiffness. In terms of practical implications, our findings suggest that regular incorporation of stretching exercises into daily routines could be beneficial in improving or maintaining trunk flexibility, which in turn could contribute to improved vascular health. Given the association between lack of trunk flexibility and increased arterial stiffness, promoting flexibility through specific exercises could be a simple but effective strategy to reduce cardiovascular risk. Future intervention studies are needed to confirm the potential benefits of flexibility training in this context and to establish specific guidelines for incorporating these exercises into cardiovascular health programmes.

## Supporting information

**S1 Checklist. MOOSE checklist.**
(DOC)

**S1 File. Supplementary tables and figures.**
(DOCX)

**S2 File. Data availability.**
(DOCX)

## Author Contributions

**Conceptualization:** Iván Cavero-Redondo.

**Data curation:** Helder Fonseca.

**Formal analysis:** Iván Cavero-Redondo, Alicia Saz-Lara.

**Funding acquisition:** Helder Fonseca.

**Investigation:** Iván Cavero-Redondo, Carla Geovanna Lever-Megina.

**Methodology:** Carla Geovanna Lever-Megina.

**Project administration:** Helder Fonseca.

**Resources:** Iris Otero-Luis, Lucimere Bohn.

**Software:** Lucimere Bohn.

**Supervision:** Iris Otero-Luis, Alicia Saz-Lara.

**Validation:** Nerea Moreno-Herraiz, Alicia Saz-Lara.

**Visualization:** Carla Geovanna Lever-Megina, Alicia Saz-Lara.

**Writing – original draft:** Iván Cavero-Redondo.

**Writing – review & editing:** Iris Otero-Luis, Nerea Moreno-Herraiz.

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
