## [Decision Letter · Decision Letter 0]

16 Aug 2024

PONE-D-24-32570Exploring the Relationship Between Trunk Flexibility and Arterial Stiffness Measured by Pulse Wave Velocity: a systematic review and meta-analysisPLOS ONE

Dear Dr. Otero Luis,

Thank you for submitting your manuscript to PLOS ONE. After careful consideration, we feel that it has merit but does not fully meet PLOS ONE’s publication criteria as it currently stands. Therefore, we invite you to submit a revised version of the manuscript that addresses the points raised during the review process.

We look forward to receiving your revised manuscript.

Kind regards,

Sebastian Schnaubelt, MD, PhD

Academic Editor

PLOS ONE

Journal Requirements:

- https://pubs.rsc.org/en/content/articlelanding/2024/fo/d3fo05061k

In your revision ensure you cite all your sources (including your own works), and quote or rephrase any duplicated text outside the methods section. Further consideration is dependent on these concerns being addressed.

Reviewers' comments:

Reviewer's Responses to Questions

**Comments to the Author**

1. Is the manuscript technically sound, and do the data support the conclusions?

Reviewer #1: Yes

Reviewer #2: Yes

2. Has the statistical analysis been performed appropriately and rigorously? 

Reviewer #1: Yes

Reviewer #2: Yes

3. Have the authors made all data underlying the findings in their manuscript fully available?

Reviewer #1: Yes

Reviewer #2: Yes

4. Is the manuscript presented in an intelligible fashion and written in standard English?

Reviewer #1: Yes

Reviewer #2: Yes

5. Review Comments to the Author

Reviewer #1: Thank you very much for this interesting work. The authors of the present meta-analysis assess the association between trunk flexibility and arterial stiffness in different subgroups.

However, some points are unclear to me:

- Do you have any information on the general physical condition or activity levels of the participants? This information would be interesting for assessing the appropriateness of trunk flexibility as a surrogate for or its independance of the physical activity level.

- You mention in your limitation section differences in the poor and high trunk flexibility classifications. (Line 223) In my opinion it is necessary to describe these different defintions in more detail at that point to communicate what excatly has to be considered when interpreting the results cautiously. Furthermore, do or did you have access to the actual values of the participants or rely soly on the assigned categories?

Reviewer #2: Thank you very much for your work on this systematic review.

This is a systematic review of etiology and risk. The exposure is trunk flexibility measured by the sit-and-reach test, and the outcome is arterial stiffness measured by pulse wave velocity. The methodology of the paper is thorough and transparent. The authors provided a pre-published protocol in PROSPERO and a completed PRISMA checklist. The manuscript is well organized. The authors claim that there is an association between trunk flexibility and arterial stiffness. However, adding more clinical context would be beneficial for readers:

Introduction:

1. The clinical use and description of measurements (sit-and-reach test, brachial-to-ankle PWV, carotid-to-femoral PWV) might be beneficial for non-expert readers.

Methods:

1. Lines 65-66. The JBI Manual on systematic reviews of etiology and risk should also be cited, as there is no Cochrane Handbook for this type of systematic review: Moola S, Munn Z, Tufanaru C, Aromataris E, Sears K, Sfetcu R, Currie M, Lisy K, Qureshi R, Mattis P, Mu P. Systematic reviews of etiology and risk (2020). Aromataris E, Lockwood C, Porritt K, Pilla B, Jordan Z, editors. JBI Manual for Evidence Synthesis. JBI; 2024. Available from: https://synthesismanual.jbi.global. https://doi.org/10.46658/JBIMES-24-06

2. Lines 151-157. Explaining the cut-off values and clinical significance of the measurements in the Methods section might be beneficial for non-expert readers. Which values of PWV correspond to clinically significant arterial stiffness?

3. Lines 88-89: "study design: cross-sectional or baseline data from longitudinal studies." Please explain the choice of these study designs.

Discussion:

1. The limitations of the measurements and differences in results between the two types of PWV described previously in literature could be interesting.

2. Lines 222-223: “This heterogeneity may be attributed to differences in participant characteristics or measurement techniques. Second, there were differences in the poor and high trunk flexibility classifications.” The authors conducted subgroup and sensitivity analyses by age, measurements, and sex, and also discussed the effects of each of these variables on the differences in results. Therefore, this limitation might be explained by these analyses and discussions unless there are additional participant characteristics contributing to the heterogeneity. If so, please elaborate on these characteristics.

3. Lines 227-228. Please elaborate on the implications for further research—for example, which possible confounders should be controlled.

Conclusion:

1. Practical implications, such as the role of stretching exercises mentioned in the discussion, might be of interest.

6. PLOS authors have the option to publish the peer review history of their article (what does this mean?). If published, this will include your full peer review and any attached files.

Reviewer #1: No

Reviewer #2: No

---

## [Author Response · Author response to Decision Letter 0]

19 Aug 2024

Dear Editor,

PLOS ONE

Enclosed you will find a revision of our manuscript: Exploring the Relationship Between Trunk Flexibility and Arterial Stiffness Measured by Pulse Wave Velocity: a systematic review and meta-analysis. Manuscript ID: PONE-D-24-32570.

We would like to thank you for giving us the opportunity to revise and improve our manuscript; we also thank the reviewers for their thoughtful and constructive comments. 

We have considered all of the suggestions and have incorporated them into the revised manuscript. Changes to the original manuscript are marked in red. We believe our manuscript is stronger as a result of these modifications. An itemised point-by-point response to the editor and reviewers’ comments is presented below.

Iris Otero-Luis

Universidad de Castilla-La Mancha

E-mail: iris.otero@uclm.es

Telephone: +34926053828

Reviewer #1: Thank you very much for this interesting work. The authors of the present meta-analysis assess the association between trunk flexibility and arterial stiffness in different subgroups.

Authors:

Thank you for the reviewer's comment. We greatly appreciate the reviewer´s time in reviewing the manuscript.

However, some points are unclear to me:

- Do you have any information on the general physical condition or activity levels of the participants? This information would be interesting for assessing the appropriateness of trunk flexibility as a surrogate for or its independence of the physical activity level.

Authors:

Thank you for the reviewer’s insightful comment. The original studies included in our meta-analysis focus mainly on the relationship between trunk flexibility and arterial stiffness, with little detailed information on participants' general fitness or activity levels beyond the flexibility measures themselves. However, several studies (e.g. Yamamoto et al.) mention that regular physical activity, which usually includes flexibility exercises, may influence flexibility and, by extension, arterial stiffness. Unfortunately, specific data on overall physical activity levels were not available in all included studies to comprehensively assess the independence of trunk flexibility from physical activity levels. Given these limitations, we acknowledge that the analysis could benefit from additional studies in which general physical activity is more rigorously monitored or measured alongside assessments of flexibility. We have included this consideration as a limitation in our discussion to guide future research directions.

“[…]. Fifth, the lack of detailed information on the overall physical activity levels of participants in the included studies. Although flexibility is a component of fitness, often correlated with other aspects of physical activity, the independence of trunk flexibility from overall activity levels remains unclear (44). This gap suggests that further research is needed to examine whether trunk flexibility may be an independent predictor of arterial stiffness or whether it is significantly confounded by overall physical activity. Future studies should aim to include comprehensive assessments of physical activity to better understand its role in the observed associations.”

Reference:

44. Nuzzo JL. The Case for Retiring Flexibility as a Major Component of Physical Fitness. Sports Med. 2020 May;50(5):853-870.

- You mention in your limitation section differences in the poor and high trunk flexibility classifications. (Line 223) In my opinion it is necessary to describe these different definitions in more detail at that point to communicate what exactly has to be considered when interpreting the results cautiously. Furthermore, do or did you have access to the actual values of the participants or rely only on the assigned categories?

Authors:

Thank you for the reviewer’s insightful feedback. We agree that providing more detail about the differences in the definitions of poor and high trunk flexibility is essential for interpreting the results accurately. In response to the reviewer’s suggestion, we have elaborated on these differences in the limitations section. Additionally, we clarify that our analysis relied on the categories assigned by the original studies, rather than direct access to the individual participants' values.

“[…]. Second, there were differences in the poor and high trunk flexibility classifications in the included studies. In particular, some studies defined these categories based on population-specific percentiles, while others used absolute cut-off values derived from the distribution of flexibility measures within their samples. These different definitions could introduce variability in the pooled analysis, as what constitutes ‘low’ or ‘high’ flexibility could differ between study populations. Furthermore, it is important to note that our meta-analysis was based on the flexibility categories reported by the original studies, and that we did not have direct access to the raw flexibility measures of individual participants. Therefore, caution should be taken when interpreting the results, as variability in definitions could affect the generalisability of our findings. […]”

Reviewer #2: Thank you very much for your work on this systematic review.

This is a systematic review of etiology and risk. The exposure is trunk flexibility measured by the sit-and-reach test, and the outcome is arterial stiffness measured by pulse wave velocity. The methodology of the paper is thorough and transparent. The authors provided a pre-published protocol in PROSPERO and a completed PRISMA checklist. The manuscript is well organized. The authors claim that there is an association between trunk flexibility and arterial stiffness. However, adding more clinical context would be beneficial for readers:

Authors

Thank you for the reviewer's comment. We greatly appreciate the reviewer´s time in reviewing the manuscript.

Introduction:

1. The clinical use and description of measurements (sit-and-reach test, brachial-to-ankle PWV, carotid-to-femoral PWV) might be beneficial for non-expert readers.

Authors:

Thank you for the reviewer's suggestion. We agree that providing a brief description of the clinical use and relevance of the sit-and-reach test, brachial-ankle pulse wave velocity (baPWV) and carotid-femoral pulse wave velocity (cfPWV) would improve the clarity of the manuscript for non-expert readers. We have incorporated these explanations in the Introduction section to ensure that all readers can better understand the importance of these measurements in the context of our study.

“The sit-and-reach test is a simple and widely used method to assess trunk flexibility, particularly the flexibility of the lower back and hamstring muscles. It is commonly used in both clinical and fitness settings to evaluate an individual's range of motion and is considered a general indicator of overall flexibility (16). Furthermore, pulse wave velocity (PWV) is a non-invasive measure of arterial stiffness, which is a key indicator of vascular health (17). Specifically, brachial-to-ankle PWV (baPWV) measures the stiffness of peripheral arteries, while carotid-to-femoral PWV (cfPWV) focuses on central arterial stiffness, particularly the aorta (18). These measurements are clinically relevant as higher PWV values are associated with increased cardiovascular risk and are used to predict adverse cardiovascular events (19-21).”

References:

16. Lemmink KA, Kemper HC, de Greef MH, Rispens P, Stevens M. The validity of the sit-and-reach test and the modified sit-and-reach test in middle-aged to older men and women. Res Q Exerc Sport. 2003 Sep;74(3):331-6.

17. Cavero-Redondo I, Sequí-Domínguez I, Saz-Lara A, Garcia-Klepzig JL, Lucerón-Lucas-Torres M, Martínez-García I, Álvarez-Bueno C, Martínez-Vizcaíno V. Concordance among pulse wave velocity assessment methods: A network meta-analysis. Chinese Medical Journal. 2024 ():10.1097/CM9.0000000000003205.

18. Tanaka H, Munakata M, Kawano Y, Ohishi M, Shoji T, Sugawara J, Tomiyama H, Yamashina A, Yasuda H, Sawayama T, Ozawa T. Comparison between carotid-femoral and brachial-ankle pulse wave velocity as measures of arterial stiffness. J Hypertens. 2009 Oct;27(10):2022-7.

19. Sequí-Domínguez I, Cavero-Redondo I, Álvarez-Bueno C, Pozuelo-Carrascosa DP, Nuñez de Arenas-Arroyo S, Martínez-Vizcaíno V. Accuracy of Pulse Wave Velocity Predicting Cardiovascular and All-Cause Mortality. A Systematic Review and Meta-Analysis. J Clin Med. 2020 Jul 2;9(7):2080.

20. Ohkuma T, Ninomiya T, Tomiyama H, Kario K, Hoshide S, Kita Y, Inoguchi T, Maeda Y, Kohara K, Tabara Y, Nakamura M, Ohkubo T, Watada H, Munakata M, Ohishi M, Ito N, Nakamura M, Shoji T, Vlachopoulos C, Yamashina A; Collaborative Group for J-BAVEL (Japan Brachial-Ankle Pulse Wave Velocity Individual Participant Data Meta-Analysis of Prospective Studies)*. Brachial-Ankle Pulse Wave Velocity and the Risk Prediction of Cardiovascular Disease: An Individual Participant Data Meta-Analysis. Hypertension. 2017 Jun;69(6):1045-1052.

21. Ben-Shlomo Y, Spears M, Boustred C, May M, Anderson SG, Benjamin EJ, Boutouyrie P, Cameron J, Chen CH, Cruickshank JK, Hwang SJ, Lakatta EG, Laurent S, Maldonado J, Mitchell GF, Najjar SS, Newman AB, Ohishi M, Pannier B, Pereira T, Vasan RS, Shokawa T, Sutton-Tyrell K, Verbeke F, Wang KL, Webb DJ, Willum Hansen T, Zoungas S, McEniery CM, Cockcroft JR, Wilkinson IB. Aortic pulse wave velocity improves cardiovascular event prediction: an individual participant meta-analysis of prospective observational data from 17,635 subjects. J Am Coll Cardiol. 2014 Feb 25;63(7):636-646.

Methods:

1. Lines 65-66. The JBI Manual on systematic reviews of etiology and risk should also be cited, as there is no Cochrane Handbook for this type of systematic review: Moola S, Munn Z, Tufanaru C, Aromataris E, Sears K, Sfetcu R, Currie M, Lisy K, Qureshi R, Mattis P, Mu P. Systematic reviews of etiology and risk (2020). Aromataris E, Lockwood C, Porritt K, Pilla B, Jordan Z, editors. JBI Manual for Evidence Synthesis. JBI; 2024. Available from: https://synthesismanual.jbi.global. https://doi.org/10.46658/JBIMES-24-06

Authors:

Thank you to the reviewer for bringing this issue to our attention. We agree that the JBI Handbook on Systematic Reviews of Aetiology and Risk is the appropriate reference for this type of review. We have cited the JBI Handbook in the Methods section, replacing the reference to the Cochrane Handbook. This ensures that our methodology is correctly aligned with the guidelines relevant to the aim of our study.

“In conducting this systematic review, we adhered to the guidelines set forth by the MOOSE statement to ensure comprehensive and accurate data synthesis (22). Additionally, this systematic review was conducted following the guidelines outlined in the JBI Manual for Evidence Synthesis on systematic reviews of etiology and risk (23). […]”

Reference:

23. Moola S, Munn Z, Tufanaru C, Aromataris E, Sears K, Sfetcu R, Currie M, Lisy K, Qureshi R, Mattis P, Mu P. Systematic reviews of etiology and risk. In: Aromataris E, Lockwood C, Porritt K, Pilla B, Jordan Z, editors. JBI Manual for Evidence Synthesis. JBI; 2024. Available from: https://synthesismanual.jbi.global.

2. Lines 151-157. Explaining the cut-off values and clinical significance of the measurements in the Methods section might be beneficial for non-expert readers. Which values of PWV correspond to clinically significant arterial stiffness?

Authors:

Thank you for the reviewer's valuable suggestion. We agree that providing an explanation of the cut-off values and their clinical significance for PWV measurements would improve the clarity of the Methods section for non-expert readers. We have now included this information to ensure that readers can better understand the implications of the PWV values used in our analysis.

“[…]; (4) outcome: arterial stiffness measured by baPWV or cfPWV. PWV is a widely recognised measure of arterial stiffness, with higher values indicating stiffer arteries and increased cardiovascular risk (25). In the case of cfPWV, a value above 1000 cm/s is generally considered a threshold of significant arterial stiffness, which is associated with an increased risk of cardiovascular events such as myocardial infarction and stroke (26). In contrast, for baPWV, a threshold of approximately 1400 cm/s is generally considered indicative of significant arterial stiffness (27); […]”

References:

25. Mitchell GF, Hwang SJ, Vasan RS, Larson MG, Pencina MJ, Hamburg NM, Vita JA, Levy D, Benjamin EJ. Arterial stiffness and cardiovascular events: the Framingham Heart Study. Circulation. 2010 Feb 2;121(4):505-11.

26. Greve SV, Blicher MK, Kruger R, Sehestedt T, Gram-Kampmann E, Rasmussen S, Vishram JK, Boutouyrie P, Laurent S, Olsen MH. Estimated carotid-femoral pulse wave velocity has similar predictive value as measured carotid-femoral pulse wave velocity. J Hypertens. 2016 Jul;34(7):1279-89.

27. Yamashina A, Tomiyama H, Arai T, Hirose K, Koji Y, Hirayama Y, Yamamoto Y, Hori S. Brachial-ankle pulse wave velocity as a marker of atherosclerotic vascular damage and cardiovascular risk. Hypertens Res. 2003 Aug;26(8):615-22.

3. Lines 88-89: "study design: cross-sectional or baseline data from longitudinal studies." Please explain the choice of these study designs.

Authors:

Thank you for the reviewer's comment. We recognise the importance of clarifying the rationale for choosing to include cross-sectional studies and baseline data from longitudinal studies in our analysis. We have added an explanation to the Methods section to better justify this choice and provide readers with a clearer understanding of our choice of study design.

“[…]; and (5) study design: cross-sectional or baseline data from longitudinal studies. The reason to include cross-sectional studies and baseline data from longitudinal studies was based on the objective of assessing the relationship between trunk flexibility and arterial stiffness at a specific point in time in various populations. Cross-sectional studies are ideally suited to capture this relationship at a single point in time, providing a snapshot of how these variables correlate in different demographic groups (28). The inclusion of baseline data from longitudinal studies allows the integration of results from studies that, while designed to track changes over time, also provide valuable baseline data comparable to cross-sectional studies. This approach allows for a more comprehensive analysis of existing evidence while maintaining consistency in the temporal assessment of the relationship between trunk flexibility and arterial stiffness. […]”

Reference:

28. Kesmodel US. Cross-sectional studies - what are they good for? Acta Obstet Gynecol Scand. 2018 Apr;97(4):388-393.

Discussion:

1. The limitations of the measurements and differences in results between the two types of PWV described previously in literature could be interesting.

Authors:

Thank you for the reviewer's suggestion. We agree that discussing the limitations of the different measurements of PWV (cfPWV and baPWV) and the possible differences in results between these two types, as described in the literature above, would add valuable context to our discussion. We have now incorporated this discussion into the manuscript to provide a more complete analysis of the findings.

“An important limitation to consider is the inherent differences between cfPWV and baPWV measurements. cfPWV is often considered the gold standard for assessing central arterial stiffness, as it directly measures aortic stiffness, which is closely related to cardiovascular events (39). In contrast, baPWV captures peripheral arterial stiffness, which may have different clinical implications (40). Previous studies have highlighted that baPWV values are often higher than cfPWV, due to the inclusion of peripheral arteries in the measurement (18). These differences could influence the interpretation of arterial stiffness and associated cardiovascular risk. Consequently, when comparing or interpreting results, it is crucial to consider these methodological differences to avoid overestimation or misclassification of vascular health status. Future studies should aim to standardise measurements of PWV or adjust for these differences to ensure consis

---

## [Decision Letter · Decision Letter 1]

23 Sep 2024

Exploring the Relationship Between Trunk Flexibility and Arterial Stiffness Measured by Pulse Wave Velocity: a systematic review and meta-analysis

PONE-D-24-32570R1

Dear Dr. Otero Luis,

We’re pleased to inform you that your manuscript has been judged scientifically suitable for publication and will be formally accepted for publication once it meets all outstanding technical requirements.

Kind regards,

Sebastian Schnaubelt, MD, PhD

Academic Editor

PLOS ONE

Additional Editor Comments (optional):

Reviewers' comments:

Reviewer's Responses to Questions

**Comments to the Author**

1. If the authors have adequately addressed your comments raised in a previous round of review and you feel that this manuscript is now acceptable for publication, you may indicate that here to bypass the “Comments to the Author” section, enter your conflict of interest statement in the “Confidential to Editor” section, and submit your "Accept" recommendation.

Reviewer #2: (No Response)

2. Is the manuscript technically sound, and do the data support the conclusions?

Reviewer #2: (No Response)

3. Has the statistical analysis been performed appropriately and rigorously? 

Reviewer #2: (No Response)

4. Have the authors made all data underlying the findings in their manuscript fully available?

Reviewer #2: (No Response)

5. Is the manuscript presented in an intelligible fashion and written in standard English?

Reviewer #2: (No Response)

6. Review Comments to the Author

Reviewer #2: (No Response)

7. PLOS authors have the option to publish the peer review history of their article (what does this mean?). If published, this will include your full peer review and any attached files.

Reviewer #2: No

---

## [Editor Report · Acceptance letter]

9 Oct 2024

PONE-D-24-32570R1 

PLOS ONE

Dear Dr. Otero Luis, 

I'm pleased to inform you that your manuscript has been deemed suitable for publication in PLOS ONE. Congratulations! Your manuscript is now being handed over to our production team.

Kind regards, 

on behalf of

Dr. Sebastian Schnaubelt 

Academic Editor

PLOS ONE